# Fosfomycin Resistance in *Escherichia coli* Isolates from South Korea and *in vitro* Activity of Fosfomycin Alone and in Combination with Other Antibiotics

**DOI:** 10.3390/antibiotics9030112

**Published:** 2020-03-06

**Authors:** Hyeri Seok, Ji Young Choi, Yu Mi Wi, Dae Won Park, Kyong Ran Peck, Kwan Soo Ko

**Affiliations:** 1Division of Infectious Diseases, Department of Medicine, Korea University Ansan Hospital, Korea University College of Medicine, Ansan 15355, Korea; hyeri.seok@gmail.com (H.S.); pugae1@hanmail.net (D.W.P.); 2Department of Microbiology, Sungkyunkwan University School of Medicine, Suwon 16419, Korea; choiji02@hotmail.com; 3Division of Infectious Diseases, Samsung Changwon Hospital, Sungkyunkwan University School of Medicine, Changwon 51353, Korea; yrhg95@naver.com; 4Division of Infectious Diseases, Samsung Medical Center, Sungkyunkwan University School of Medicine, Seoul 06351, Korea; krpeck@skku.edu

**Keywords:** fosfomycin, in vitro time-kill, cefixime, piperacillin-tazobactam

## Abstract

We investigated fosfomycin susceptibility in *Escherichia coli* clinical isolates from South Korea, including community-onset, hospital-onset, and long-term care facility (LTCF)-onset isolates. The resistance mechanisms and genotypes of fosfomycin-resistant isolates were also identified. Finally, the in vitro efficacy of combinations of fosfomycin with other antibiotics were examined in susceptible or extended spectrum β-lactamase (ESBL)-producing *E. coli* isolates. The fosfomycin resistance rate was 6.7% and was significantly higher in LTCF-onset isolates than community-onset and hospital-onset isolates. Twenty-one sequence types (STs) were identified among 19 fosfomycin-resistant *E. coli* isolates, showing diverse genotypes. *fosA3* was found in only two isolates, and diverse genetic variations were identified in three genes associated with fosfomycin resistance, namely, GlpT, UhpT, and MurA. Some fosfomycin-resistant *E. coli* isolates carried no mutations. In vitro time-kill assays showed that fosfomycin alone did not exhibit an excellent killing activity, compared with ciprofloxacin in susceptible isolates and with ertapenem in ESBL producers. However, combining fosfomycin with cefixime or piperacillin-tazobactam eradicated susceptible or ESBL-producing isolates, respectively, even with 0.5× minimum inhibitory concentrations. Overall, we found a relatively high fosfomycin resistance rate in *E. coli* isolates from South Korea. Based on their genotypes and resistance mechanisms, most of the fosfomycin-resistant *E. coli* isolates might occur independently. Antibiotic combinations with fosfomycin could be a suitable therapeutic option for infections caused by *E. coli* isolates.

## 1. Introduction

*Escherichia coli* is one of the most common pathogens in community-acquired and nosocomial infections, including urinary tract infections (UTIs), biliary tract infections, and complicated intraabdominal infections. Indeed, one in every three adult women suffers from UTIs and about 50% of UTIs are caused by *E. coli* [1,2]. Although *E. coli* is intrinsically susceptible to many antimicrobial agents, antimicrobial resistance has been increasingly reported due to extended-spectrum β-lactamases (ESBLs), carbapenemases, plasmid-mediated quinolone resistance, and *mcr* genes causing colistin resistance [3]. The increase of antimicrobial resistance in *E. coli* leads to the reduction of usable therapeutic agents and prolongs the length of hospital stay due to the absence of effective oral antibiotics.

Fosfomycin, an old antimicrobial drug, has resurfaced as a therapeutic option for multidrug-resistant (MDR) gram-negative bacilli [4]. Its mechanism of action is to inhibit the formation of the peptidoglycan precursor UDP N-acetylmuramic acid (UDP-MurNAc) in the bacterial cell wall biosynthesis [5]. Since fosfomycin is structurally unrelated to any other antimicrobial agent, there is a small chance of cross-resistance [6]. Additionally, it has a broad-spectrum activity against both gram-negative and gram-positive bacteria, with limited side effects [7,8]. It is known that *E. coli* has two main nutrient transport systems essential for fosfomycin uptake: the glycerol-3-phosphate transporter (GlpT) and a hexose phosphate transporter, known as the glucose-6-phosphate transporter (UhpT) [9]. Thus, the key fosfomycin resistance mechanisms involve reduced permeability related to GlpT and UhpT, and target modification related to MurA. In addition, drug inactivation can be caused by the acquisition of *fos* genes mostly by plasmid, resulting in fosfomycin resistance [6,8,10]. 

The current resistance rate of *E. coli* to fosfomycin is estimated to be lower than 5%, and lower than 10% among extended spectrum β-lactamase (ESBL) producers worldwide [11,12,13]. In South Korea, the fosfomycin resistance rate has been reported to be up to 3% in clinical *E. coli* isolates, and up to 7% among ESBL producers [14,15,16,17]. However, only a few studies have explored the mechanisms of fosfomycin resistance and the genotypes of fosfomycin-resistant *E. coli* isolates in South Korea.

In this study, we investigated the antimicrobial susceptibility of different *E. coli* clinical isolates from South Korea, including community-onset, hospital-onset, and long-term care facility (LTCF)-onset isolates. We also identified the resistance mechanisms and genotypes of fosfomycin-resistant isolates. In addition, the in vitro efficacy of combinations of fosfomycin with other antibiotics were examined in *E. coli* isolates, including ESBL producers. Although combination therapy based on fosfomycin is not commonly used, we explored the possibilities of combination therapy, especially using oral antibiotics such as ciprofloxacin and cefixime.

## 2. Results

Among the 283 *E. coli* clinical isolates from South Korea, 19 isolates (6.7%) were resistant to fosfomycin based on the cut-off minimum inhibition concentration (MIC) (Table 1). The fosfomycin resistance rates were higher in LTCF-onset isolates (10.0%) than in community-onset and hospital-onset isolates (6.0% and 6.9%, respectively), but their difference was not statistically significant. The difference of fosfomycin resistant rates between hospitals was also not statistically significant, although it was significantly lower in urinary tract infection (UTI) isolates than in non-UTI isolates (*P*, 0.042) (Table 1). 

The ciprofloxacin resistance rate was very high (64.0% in all isolates), especially in UTI isolates and isolates from KUAH (Table 1). Isolates resistant to cephalosporin (cefepime and cefixime) were found more frequently in hospital- and LTCF-onset isolates. The piperacillin–tazobactam resistance rate was higher in isolates from SMC and SCH, unlike fosfomycin, ciprofloxacin, and cefepime resistance rates. The amikacin resistance rate was very low (2.1%), and only seven ertapenem-resistant isolates were identified (2.5%). The colistin resistance rate was 10.6%. While the colistin resistance rate did not differ in the mode of acquisition and site of infection, it was significantly higher in isolates from SMC (17.5%) compared to those from the other two hospitals (6.5% in SCH and 4.8% in KUAH, respectively). Only three isolates were resistant to tigecycline. Multidrug resistance (MDR), defined as resistance to ≥3 antibiotic classes, was identified in 116 isolates (41.0%).

Using the MLST analysis, the genotypes of 19 fosfomycin-resistant *E. coli* isolates were identified (Table 2). A total of 14 STs were identified. Only three of these STs were found in multiple isolates: ST1193 in four isolates, while ST131 and ST1531-slv in two isolates each. Eight STs were newly identified, and they were defined as slv or dlv of the most closely related ST. The largest CCs were CC131 and CC14. All CC131 isolates except one (C072) were MDR. The fosfomycin resistance rates were similar between isolates from blood (16/243 isolates, 6.6%) and urine (3/40 isolates, 7.5%). 

As for fosfomycin resistance mechanisms, *fosA3* was identified in only two isolates (A004 and C063) (Table 2). Amino acid substitutions or insertions in GlpT, UhpT, and MurA were found in eight, one, and two fosfomycin-resistant isolates, respectively. Only one mutation, A16T in GlpT was identified in multiple fosfomycin-resistant *E. coli* isolates belonging to the same genotype (ST1531-slv). Only one isolate (C063) contained both *fosA3* and amino acid alterations in GlpT and MurA. Neither *fosA3* nor amino acid alterations in three genes were identified in eight fosfomycin-resistant isolates. 

To investigate the antimicrobial effects of fosfomycin and other antibiotics with in vitro time-kill assays, we selected four *E. coli* isolates susceptible to fosfomycin, two susceptible to ciprofloxacin and cefixime, and two resistant to ciprofloxacin and cefixime and producing ESBLs (Table 3). These belonged to different STs, namely, ST216, ST144-slv, ST131, and ST1193. Two isolates producing ESBLs were identified (A038 and C047), producing CTX-M-15 and CTX-M-14, respectively.

While 1× and 0.5× MICs of cefixime and fosfomycin did not eradicate both susceptible *E. coli* isolates (C093 and S088), 1× MIC of ciprofloxacin showed complete killing efficacy after 4 or 8 h (Figure 1A,B). Using 0.5× MIC of ciprofloxacin also killed the isolates, but the killing efficacy was lower than with 1× MIC. The combination of 0.5× MICs of fosfomycin and other antibiotics (ciprofloxacin or cefixime) eradicated the isolates (Figure 1C,D). Interestingly, the combination of 0.5× MICs of fosfomycin and cefixime showed a synergistic killing effect; while single regimens of each antibiotic did not kill the susceptible isolates, using a combination of the two completely eradicated the isolates. 

For the two ESBL-producing *E. coli* isolates, ertapenem and piperacillin-tazobactam were also tested in addition to fosfomycin (Figure 2). While 1× MIC of ertapenem completely killed both ESBL-producing isolates, 1× MIC of fosfomycin decreased the growth of both isolates within 2 or 4 h, after which they started growing again (Figure 2A,B). In contrast, 1× MIC of piperacillin-tazobactam produced different results between the two ESBL-producing isolates. Although the combinations of 0.5× MICs of fosfomycin with ertapenem or piperacillin-tazobactam completely eradicated both ESBL-producing isolates after 24 h, 0.5x MIC of ertapenem and fosfomycin also completely killed them after 4 h (Figure 2C,D). The combinations of 0.5× MICs of fosfomycin and ertapenem or piperacillin-tazobactam completely eradicated both ESBL-producing isolates after 24 h. 

## 3. Discussion

In this study, the fosfomycin resistance rate was estimated to be 6.7%, which is somewhat higher than that previously reported in South Korea [14,15,16,17]. This study identified different genotypes among 19 fosfomycin-resistant *E. coli* isolates, suggesting that most of these isolates might emerge sporadically. The most prevalent clonal group in these isolates was CC131 and CC14. It is known that ST131 is tightly associated with the production of the CTX-M-type ESBL and with fluoroquinolone resistance, and is therefore commonly acknowledged as a significant threat to public health [18,19,20]. The fosfomycin-resistant *E. coli* CC131 isolates in this study were all MDR, except for one. Since the genetic alterations of fosfomycin resistance-associated genes were not identical among the CC131 isolates, it does not appear that all fosfomycin resistant strains belonging to CC131 have spread clonally. In addition, ST1193, found in four fosfomycin-resistant isolates, may not be spread clonally. Although it is still unclear whether fosfomycin resistance preferably occurs in certain genetic backgrounds, it is worth investigating whether ST131 or ST1193 increase the risk of developing fosfomycin resistance. 

We investigated the presence of genetic mutations in several genes (*fosA*, *fosC*, *glpT*, *uhpT*, and *murA)* that are commonly associated with fosfomycin resistance [6,21]. Diverse genetic alterations in these genes were found, including the insertion of two amino acids in GlpT. This supports the idea that fosfomycin resistance in *E. coli* isolates might occur independently between each other. Additionally, the fosfomycin-modifying enzyme FosA3, a metalloenzyme acquired through plasmid-transfer [9], was identified in only two isolates, although the *fosA* gene has been reported to be prevalent in Asia [22]. This suggests that the horizontal transfer of fosfomycin resistance genes through mobile elements is not common in South Korea. However, no genetic alterations in GlpT, UhpT, and MurA were found in many isolates, particularly isolates belonging to CC131. Further fosfomycin resistance mutations reported in other studies, such as those in *cyaA* and *ptsI* resulting in lower cAMP levels and downregulation of fosfomycin transporters [23,24], may be associated to some fosfomycin-resistant *E. coli* isolates. Further unknown mechanisms might also mediate fosfomycin resistance in *E. coli*.

Some *E. coli* studies have shown a synergistic effect of fosfomycin in combination with other antibiotics, such as carbapenems, colistin, aztreonam, netilmicin, and tigecycline, against ESBL-producing strains [9]. In this study, the activity of antibiotic combinations based on fosfomycin were evaluated by in vitro time-kill assay using multiple *E. coli* isolates. Different antibiotic combinations were tested according to antibiotic susceptibility. For both antibiotic susceptible and ESBL-producing *E. coli* isolates, fosfomycin alone did not show lower killing activity than other antibiotics, such as ciprofloxacin against susceptible isolates and ertapenem against ESBL-producers. However, the combination of fosfomycin with other antibiotics, including cefixime and piperacillin-tazobactam, achieved complete bactericidal effects against susceptible isolates and ESBL-producing isolates, respectively, even using 0.5× MIC. This suggests that antibiotic combinations of fosfomycin and other antibiotics, even at low concentrations, could be a potential treatment option for infections caused by *E. coli*.

This study investigated fosfomycin resistance in clinical *E. coli* isolates from South Korea. The fosfomycin resistance rate was 6.7%, and the rates were significantly lower in UTI isolates. The fosfomycin-resistant isolates showed diverse genotypes and genetic variations in genes associated with fosfomycin resistance, indicating sporadic emergence of fosfomycin resistance in South Korea. Although fosfomycin monotherapy was not superior to other antibiotics in its killing activity against susceptible and ESBL-producing *E. coli* isolates, its combination with other antibiotics, even at low concentrations, resulted in a synergistic effect.

## 4. Materials and Methods

### 4.1. Bacterial Isolates 

A total of 283 nonduplicated *E. coli* clinical isolates (243 from blood and 40 from urine) were collected from patients from January to June 2018 from three tertiary-care hospitals in South Korea: the Samsung Medical Center (SMC, 114 isolates), the Samsung Changwon Hospital (SCH, 107 isolates), and the Korea University Ansan Hospital (KUAH, 62 isolates). Species identification was performed using a VITEK-2 system (BioMérieux, Hazelwood, MO, USA).

### 4.2. Antimicrobial Susceptibility Testing

In vitro antimicrobial susceptibility testing was performed in all *E. coli* isolates according to the Clinical and Laboratory Standard Institute (CLSI) guidelines [25]. An agar dilution method with glucose-6-phophate was used for fosfomycin, and a broth microdilution method was applied for the other eight antimicrobial agents that were tested: ciprofloxacin, cefixime, cefepime, ertapenem, piperacillin-tazobactam, amikacin, colistin, and tigecycline. *E. coli* ATCC 25922 and *Pseudomonas aeruginosa* ATCC 27853 were used as control strains. The antimicrobial susceptibility testing was performed in duplicate. 

### 4.3. Genotyping

Multilocus sequence typing (MLST) analysis was performed to determine the genotypes of fosfomycin-resistant *E. coli* isolates as previously described [26]. The sequence type (ST) and clonal complex (CC) were determined based on the database available at https://pubmlst.org/escherichia/, and those not matching exactly with STs assigned in the database were designated as single-, double-, and triple-locus variants (slv, dlv, and tlv) of the most closely related STs. 

### 4.4. Fosfomycin Resistance Mechanisms

Polymerase chain reaction (PCR) and sequencing were performed for fosfomycin-resistant *E. coli* isolates to identify the following genes associated with fosfomycin resistance: *murA*, *glpT*, *uhpT*, *fosA3*, and *fosC2*. The primers used for amplification and sequencing were based on previous studies [13,27,28]. 

### 4.5. In vitro Time-Kill Assays

The activity of fosfomycin and other antibiotics was evaluated by time-kill assays using an inoculum of 1 × 10^6^ CFU/mL of *E. coli*. Fosfomycin, ciprofloxacin, and cefixime were evaluated against two ESBL-nonproducing *E. coli* isolates (C093 and S088), while fosfomycin, ertapenem, and piperacillin-tazobactam were evaluated against ESBL-producing isolates (A038 and C046). ESBL production was identified using a double-disc synergy test and was confirmed by PCR for *bla*_CTX-M_ [29]. Antibiotic dilutions were prepared in 10 ml of PBS adjusted to final concentrations of 0.5× and 1× the MIC of the test strains. Efficacy of the combination of fosfomycin and other antibiotics was evaluated using concentrations of 0.5× MICs. PBS without antibiotics served as a growth control; PBS without *E. coli* served as a negative control. Bacterial growth was quantified after 0, 2, 4, 8, and 24 h incubation at 37 °C by plating 10-fold dilutions on sheep blood agar. Colony forming unit (CFU) counts were determined for killing curves of antibiotics. Antimicrobials were considered bactericidal when a ≥ 3 log_10_ decrease in CFU/mL was reached compared with the initial inoculate. 

### 4.6. Definition of Infection Onset

An infection occurring in the community or up to 48 h after hospital admission was defined as community-onset, whereas one occurring more than 48 h after admission was defined as hospital-onset. Patients who were referred from LTCF were considered as LTCF-onset. The site of infection was classified as urinary tract or non-urinary tract because of the heterogeneity of infection sites. 

### 4.7. Statistical Analysis 

To compare the two groups, Pearson χ² tests and Fisher’s exact tests were used for categorical variables, and student’s t-test and Mann–Whitney U tests were used for continuous variables where appropriate. All statistical tests were two-tailed, and *P* values ≤ 0.05 were considered statistically significant. 

## Figures and Tables

**Figure 1 antibiotics-09-00112-f001:**
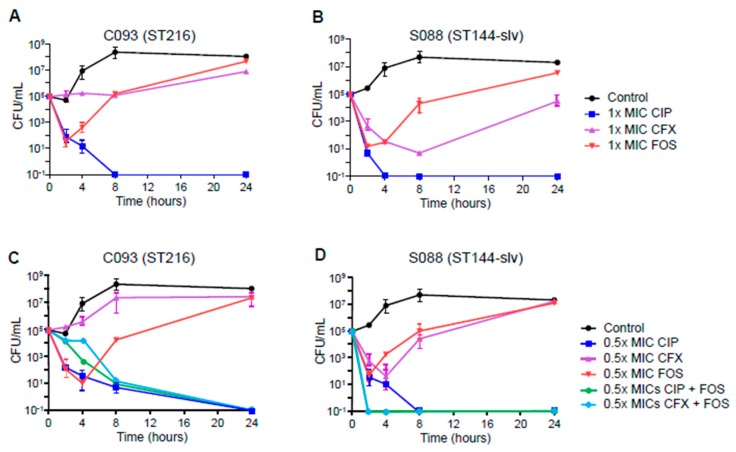
Time-kill curves for ciprofloxacin, cefixime, and fosfomycin against susceptible *E. coli* isolates, C093 (ST216) and S088 (ST144-slv). (**A** and **B**), the results of 1× MICs of single antibiotics, (**C** and **D**), the results of 0.5× MICs of single and combination of antibiotics. CIP, ciprofloxacin; CFX, cefixime; FOS, fosfomycin.

**Figure 2 antibiotics-09-00112-f002:**
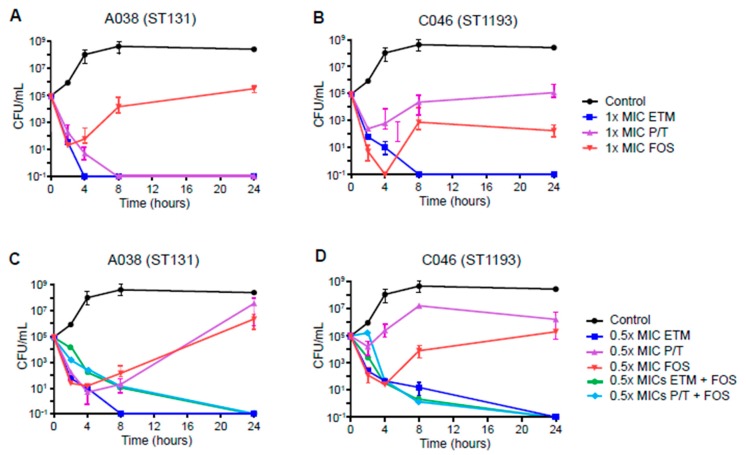
Time-kill curves for ertapenem, piperacillin-tazobactam, and fosfomycin against ESBL-producing *E. coli* isolates, CTX-M-15-producing A038 (ST131) and CTX-M-14-producing C046 (ST1193). (**A** and **B**), the results of 1× MICs of single antibiotics, (**C** and **D**), the results of 0.5× MICs of single and combination of antibiotics. ETM, ertapenem; P/T, piperacillin-tazobactam; FOS, fosfomycin.

**Table 1 antibiotics-09-00112-t001:** Antibiotics susceptibilities of all *E. coli* isolates in this study.

Antimicrobial Agents	Number of Resistant Isolates (%)											
	Total (*n* = 283)	Mode of Acquisition				Site of Infection ^c^			Facility ^e^			
		Community-onset (*n =* 150)	Hospital-onset (*n* = 102)	LTCF ^b^-onset (*n* = 31)	*P*	UTI ^d^ (*n* = 129)	Non-UTI (*n* = 154)	*P*	SMC (*n* = 114)	SCH (*n* = 107)	KUAH (*n* = 62)	*P*
Fosfomycin	19 (6.7)	9 (6.0)	7 (6.9)	3 (10.0)	0.384	5 (3.9)	14 (9.1)	0.042	4 (3.5)	9 (8.4)	6 (9.7)	0.054
Ciprofloxacin	181 (64.0)	72 (48.0)	70 (68.6)	22 (71.0)	0.285	70 (72.9)	111 (59.4)	0.023	67 (58.8)	62 (57.9)	52 (83.9)	0.008
Cefepime	117 (41.3)	48 (32.0)	55 (53.9)	14 (45.2)	0.008	45 (34.9)	72 (46.8)	0.115	45 (39.5)	40 (37.4)	32 (51.6)	0.022
Cefixime	131 (46.3)	58 (38.7)	57 (55.9)	16 (51.6)	0.033	54 (48.9)	77 (50.0)	0.363	48 (42.1)	45 (42.1)	38 (61.3)	0.062
P/T ^a^	88 (31.1)	38 (25.4)	42 (41.2)	8 (25.8)	0.025	42 (32.6)	46 (29.9)	0.813	49 (43.0)	34 (31.8)	5 (8.1)	<0.001
Amikacin	6 (2.1)	0	4 (3.9)	2 (6.5)	0.017	4 (3.1)	2 (1.3)	0.534	5 (4.4)	0	1 (1.6)	0.116
Ertapenem	7 (2.5)	1 (0.7)	6 (5.9)	0	0.012	4 (3.1)	3 (1.9)	0.880	1 (0.9)	3 (2.8)	3 (4.8)	0.244
Colistin	30 (10.6)	12 (8.0)	15 (14.7)	3 (9.7)	0.219	18 (14.0)	12 (7.8)	0.121	20 (17.5)	7 (6.5)	3 (4.8)	0.007
Tigecycline	3 (1.1)	0	2 (2.0)	1 (3.2)	0.075	2 (1.5)	1 (0.6)	0.593	0	0	2 (3.2)	0.117

^a^ P/T, piperacillin/tazobactam. ^b^ LTCF, long-term care facility. ^c^ All patients accompanied by *E. coli* infection. ^d^ UTI, urinary tract infection. ^e^ SMC, Samsung Medical Center; SCH, Samsung Changwon Hospital; KUAH, Korea University Ansan Hospital.

**Table 2 antibiotics-09-00112-t002:** Characteristics of 29 fosfomycin-resistant *E. coli* isolates: genotype, clinical characteristics, and amino acid alterations in genes associated with fosfomycin resistance.

Isolate No.	CC ^a^	ST ^a^	Allele no. ^b^	Specimen	Site of Infection ^c^	Mode of Acquisition ^d^	Amino Acid Alterations			
							*fos*	GlpT	UhpT	MurA
S020	CC131	ST131	53-40-47-13-36-28-29	Blood	IAI	Hospital				
S074		ST131	53-40-47-13-36-28-29	Blood	UTI	Hospital		D220N		
C072		ST131-slv1	53-40-193-13-36-28-29	Blood	Cholangitis	Community				
C073		ST131-slv2	53-40-47-200-36-28-29	Blood	UTI	LTCF				
A011		ST131-dlv1	53-35-47-13-36-5-29	Blood	IAI	Hospital				
C025	CC14	ST1193	14-14-10-200-17-7-10	Urine	UTI	Community				
C036		ST1193	14-14-10-200-17-7-10	Blood	Cholangitis	Community		G168R		
A049		ST1193	14-14-10-200-17-7-10	Blood	UTI	Community		M136K		
S019		ST1193	14-14-10-200-17-7-10	Blood	Prostatitis	LTCF				
C078		ST1193-slv	14-40-10-200-17-7-10	Blood	UTI	Community				
C106	CC69	ST106	21-38-27-6-5-8-4	Blood	UTI	Community				
A004		ST106-dlv	21-88-27-6-5-79-4	Blood	UTI	Community	*fosA3*			
A031	CC95	ST1531-slv	37-35-19-37-17-5-26	Blood	UTI	Community		A16T		
A041		ST1531-slv	37-35-19-37-17-5-26	Blood	UTI	LTCF		A16T		
C011	CC155	ST58	64-4-16-24-8-14	Blood	Liver abscess	Community			Y60F	
C045	CC38	ST38	4-26-2-25-5-5-19	Urine	UTI	Hospital		Ins. of DG139		
C063	CC10	ST10	10-11-4-8-8-8-2	Urine	UTI	Hospital	*fosA3*	G168R		P99S
A043	CC398	ST398-slv	64-40-1-1-8-8-6	Blood	Cholangitis	Hospital				A154T
S050	CC95	ST95-slv	37-38-34-37-17-11-26	Blood	NF	Hospital		A16T		

^a^ CC, clonal complex; ST, sequence type. ^b^
*adk-fumc-gyrB-icd-mdh-purA-recA*, slv, single-locus variant; dlv, double-locus variant. The allele number different from the most closely related ST was underlined. ^c^ UTI, urinary tract infection; IAI, intraabdominal infection; NF, neutropenic fever. ^d^ LTCF, long-term care facility.

**Table 3 antibiotics-09-00112-t003:** Characteristics of *E. coli* isolates selected for time-kill assays.

Isolate No.	ST ^a^	ESBL Type ^b^	MIC (mg/L) ^c^									Site of Infection ^d^	Mode of Acquisition
			FOS	CIP	CFM	CFX	P/T	AMK	ETM	COL	TGC		
C093	ST216	-	4	0.06	0.06	0.25	16/4	4	0.06	1	2	Cholangitis	Community
S088	ST144-slv	-	8	0.06	0.06	0.06	1/4	4	0.06	0.5	1	Cholangitis	Community
A038	ST131	CTX-M-15	8	>64	>64	>64	16/4	4	0.06	1	4	UTI	Community
C046	ST1193	CTX-M-14	16	>64	>64	>64	4/4	4	0.06	1	1	UTI	Community
Control Strains													
*E. coli* ATCC 25922			0.5	0.06	0.06	0.5	2/4	1	0.06	0.5	0.06		
*P. aeruginosa* ATCC 27853			2	0.25	2	NA^e^	4/4	1	0.5	0.5	NA		

^a^ ST, sequence type. ^b^ ESBL, extended-spectrum-β-lactamase. ^c^ MIC, minimum inhibitory concentration; FOS, fosfomycin; CIP, ciprofloxacin; CFM, cefepime; CFX, cefixime; P/T, piperacillin-tazobactam; AMK, amikacin; ETM, ertapenem; COL, colistin; TGC, tigecycline. ^d^ UTI, urinary tract infection. ^e^ NA, not available.

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
