# Peer review of "Fosfomycin Resistance in *Escherichia coli* Isolates from South Korea and *in vitro* Activity of Fosfomycin Alone and in Combination with Other Antibiotics"

_antibiotics, 2020, doi:10.3390/antibiotics9030112_

Round 1

Reviewer 1 Report

This is an interesting and well-written manuscript that analyzed fosfomycin susceptibility in Escherichia coli clinical isolates from South Korea and the efficacy of combinations of fosfomycin with other antibiotics.

In recent years, the use of fosfomycin has increased spectacularly due to the considerable incidence of multidrug-resistant microorganisms for which fosfomycin is, alone or in combination, a treatment alternative. In this regard, the European Medicines Agency and the US Food and Drug Administration have open processes for reviewing the accumulated information on the use of fosfomycin from new clinical trials  and  susceptibility in vitro assays.

However, prior to further processing of the paper several points need to be clarified

-In Material and methods section

-- 4.2. In the Agar dilution assay, did the authors add G6P to the medium (25 mg/L)? as indicate  the Clinical and Laboratory Standards Institute

-In Discussion section

Among fosfomycin-resistant E. coli isolates,  fosA3 was identified in only two isolates. However, fosA3 is very prevalent in Asia. Can authors comment on this?

Author Response

  1. In Material and methods section

4.2. In the Agar dilution assay, did the authors add G6P to the medium (25 mg/L)? as indicate  the Clinical and Laboratory Standards Institute.

- Although we did not mention, we added G6P to the medium. We mentioned it in the revised manuscript.

“Agar dilution method with glucose-6-phophate was used for fosfomycin,” (Line 205 in the revised manuscript)

  1. In Discussion section

Among fosfomycin-resistant E. coli isolates, fosA3 was identified in only two isolates. However, fosA3 is very prevalent in Asia. Can authors comment on this?

- As suggested, we commented it in the revised manuscript.

“Additionally, the fosfomycin-modifying enzyme FosA3, a metalloenzyme acquired through plasmid-transfer [9], was identified in only two isolates, although the fosA gene has been reported to be prevalent in Asia [22].” (Line 162-164 in the revised manuscript)

Reviewer 2 Report

Hyeri Soek et. al. isolated E. coli strains from community, hospital and long-term care facilities and studied their susceptibilities and mechanisms to antibiotics especially fosfomycin. Authors also investigated potential treatment of drug-resistant isolates by co-administration of different antibiotics and reported significant synergistic effects. Overall the study was carefully designed and carried out. The results provided up-to-date fosfomycin-resistant E. coli strains in South Korea and have clear clinic importance. The research is suitable to be published in Antibiotics after addressing following questions which are mostly minor.

  1. Line 74. "19 isolates were resistant to fosfomycin..." Authors need to provide the definition of "resistant", e.g. a cut-off MIC or IC50 of a given antibiotic.
  2. Table 1. The term of "site of infection" is confusing. a) did the patients have infectious symptom caused by E. coli?  b) can "hospital" column be fit into "hospital-onset" (or list in a separate table)?  
  3. Table 3. The method used to determine MICs need to be described in very details in method section. Even slight change of the details will lead to big change in MIC results. If ATCC 25922 and ATCC 27853 were used as control strains, then their MICs need to be reported in the same table and, I would recommend, include the reported numbers to validate the data.
  4. Figure 1 and Figure 2. I don't see the error bars. Was it due to the standard deviation too low to be shown? If so I would like to suggest a table to include the raw data.
  5. A more general question, why did the authors pick those specific antibiotics for synergistic study at the first place? Were they commonly used in South Korea or recommended by medical doctors? This may provide more background information for general readers.

Author Response

  1. Line 74. "19 isolates were resistant to fosfomycin..." Authors need to provide the definition of "resistant", e.g. a cut-off MIC or IC50 of a given antibiotic.

- As suggested, we provide the definition of resistance.

“Among the 283 E. coli clinical isolates from South Korea, 19 isolates (6.7%) were resistant to fosfomycin based on cut-off minimum inhibition concentration (MIC) (Table 1).” (Line 86-87 in the revised manuscript)

  1. Table 1. The term of "site of infection" is confusing. a) did the patients have infectious symptom caused by E. coli?  b) can "hospital" column be fit into "hospital-onset" (or list in a separate table)?

- All patients included in this study were accompanied by E. coli infection.

- While the ‘hospital-onset’ column indicates “nosocomial infection, the ‘hospital’ in the last column indicate hospital name, or facility. To avoid confusion, we changed ‘hospital’ in the last column into ‘facility’.

  1. Table 3. The method used to determine MICs need to be described in very details in method section. Even slight change of the details will lead to big change in MIC results. If ATCC 25922 and ATCC 27853 were used as control strains, then their MICs need to be reported in the same table and, I would recommend, include the reported numbers to validate the data.

- As suggested, we showed the results of control strains, and provided the repeated number.

“. The antimicrobial susceptibility testing was performed in duplicate.” (Line 208-209 in the revised manuscript)

  1. Figure 1 and Figure 2. I don't see the error bars. Was it due to the standard deviation too low to be shown? If so I would like to suggest a table to include the raw data.

- We included error bars in the Figures.  

  1. A more general question, why did the authors pick those specific antibiotics for synergistic study at the first place? Were they commonly used in South Korea or recommended by medical doctors? This may provide more background information for general readers.

- Combination therapy based on fosfomycin is not commonly used in South Korea. We performed this research to explore the possibilities of combination therapy based on fosfomycin. Especially, we intended to explore combination therapy of oral antibiotics such as ciprofloxacin and cefixime. We added some background in the Introduction.

“Although combination therapy based on fosfomycin is not commonly used, we explored the possibilities of combination therapy, especially using oral antibiotics such as ciprofloxacin and cefixime.” (Line 81-83 in the revised manuscript)